# An intriguing failing of convolutional neural networks and the CoordConv solution

**Rosanne Liu**[1]    **Joel Lehman**[1]    **Piero Molino**[1]    **Felipe Petroski Such**[1]    **Eric Frank**[1]

**Alex Sergeev**[2]                    **Jason Yosinski**[1]

[1]Uber AI Labs, San Francisco, CA, USA    [2]Uber Technologies, Seattle, WA, USA

`{rosanne,joel.lehman,piero,felipe.such,mysterefrank,asergeev,yosinski}@uber.com`

## Abstract

Few ideas have enjoyed as large an impact on deep learning as convolution. For any problem involving pixels or spatial representations, common intuition holds that convolutional neural networks may be appropriate. In this paper we show a striking counterexample to this intuition via the seemingly trivial *coordinate transform problem*, which simply requires learning a mapping between coordinates in $(x, y)$ Cartesian space and coordinates in one-hot pixel space. Although convolutional networks would seem appropriate for this task, we show that they fail spectacularly. We demonstrate and carefully analyze the failure first on a toy problem, at which point a simple fix becomes obvious. We call this solution CoordConv, which works by giving convolution access to its own input coordinates through the use of extra coordinate channels. Without sacrificing the computational and parametric efficiency of ordinary convolution, CoordConv allows networks to learn either complete translation invariance or varying degrees of translation dependence, as required by the end task. CoordConv solves the coordinate transform problem with perfect generalization and 150 times faster with 10–100 times fewer parameters than convolution. This stark contrast raises the question: to what extent has this inability of convolution persisted insidiously inside other tasks, subtly hampering performance from within? A complete answer to this question will require further investigation, but we show preliminary evidence that swapping convolution for CoordConv can improve models on a diverse set of tasks. Using CoordConv in a GAN produced less mode collapse as the transform between high-level spatial latents and pixels becomes easier to learn. A Faster R-CNN detection model trained on MNIST detection showed 24% better IOU when using CoordConv, and in the Reinforcement Learning (RL) domain agents playing Atari games benefit significantly from the use of CoordConv layers.

## 1  Introduction

Convolutional neural networks (CNNs) [17] have enjoyed immense success as a key tool for enabling effective deep learning in a broad array of applications, like modeling natural images [36, 16], images of human faces [15], audio [33], and enabling agents to play games in domains with synthetic imagery like Atari [21]. Although straightforward CNNs excel at many tasks, in many other cases progress has been accelerated through the development of specialized layers that complement the abilities of CNNs. Detection models like Faster R-CNN [27] make use of layers to compute coordinate transforms and focus attention, spatial transformer networks [13] make use of differentiable cameras to transform data from the output of one CNN into a form more amenable to processing with another,

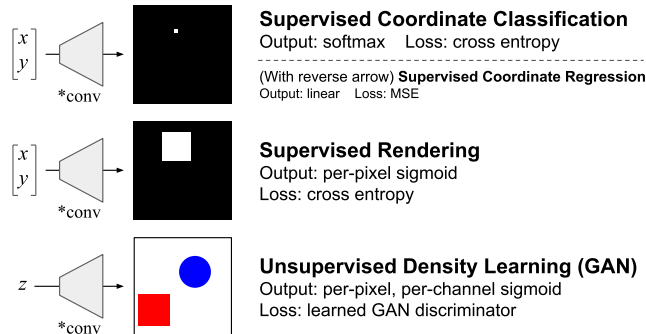

Figure 1: Toy tasks considered in this paper. The *conv* block represents a network comprised of one or more convolution, deconvolution (convolution transpose), or CoordConv layers. Experiments compare networks with no CoordConv layers to those with one or more.

and some generative models like DRAW [8] iteratively perceive, focus, and refine a canvas rather than using a single pass through a CNN to generate an image. These models were all created by neural network designers that intuited some inability or misguided inductive bias of standard CNNs and then devised a workaround.

In this work, we expose and analyze a generic inability of CNNs to transform spatial representations between two different types: from a dense Cartesian representation to a sparse, pixel-based representation or in the opposite direction. Though such transformations would seem simple for networks to learn, it turns out to be more difficult than expected, at least when models are comprised of the commonly used stacks of convolutional layers. While straightforward stacks of convolutional layers excel at tasks like image classification, they are not quite the right model for coordinate transform.

The main contributions of this paper are as follows:

1. We define a simple toy dataset, *Not-so-Clevr*, which consists of squares randomly positioned on a canvas (Section 2).

2. We define the *CoordConv* operation, which allows convolutional filters to know where they are in Cartesian space by adding extra, hard-coded input channels that contain coordinates of the data seen by the convolutional filter. The operation may be implemented via a couple extra lines of Tensorflow (Section 3).

3. Throughout the rest of the paper, we examine the coordinate transform problem starting with the simplest scenario and ending with the most complex. Although results on toy problems should generally be taken with a degree of skepticism, starting small allows us to pinpoint the issue, exploring and understanding it in detail. Later sections then show that the phenomenon observed in the toy domain indeed appears in more real-world settings.

   We begin by showing that coordinate transforms are surprisingly difficult even when the problem is *small and supervised*. In the *Supervised Coordinate Classification* task, given a pixel's $(x, y)$ coordinates as input, we train a CNN to highlight it as output. The *Supervised Coordinate Regression* task entails the inverse: given an input image containing a single white pixel, output its coordinates. We show that both problems are harder than expected using convolutional layers but become trivial by using a CoordConv layer (Section 4).

4. The *Supervised Rendering* task adds complexity to the above by requiring a network to paint a full image from the Not-so-Clevr dataset given the $(x, y)$ coordinates of the center of a square in the image. The task is still fully supervised, but as before, the task is difficult to learn for convolution and trivial for CoordConv (Section 4.3).

5. We show that replacing convolutional layers with CoordConv improves performance in a variety of tasks. On two-object Sort-of-Clevr [29] images, Generative Adversarial Networks (GANs) and Variational Autoencoders (VAEs) using CoordConv exhibit less mode collapse, perhaps because ease of learning coordinate transforms translates to ease of using latents to span a 2D Cartesian space. Larger GANs on bedroom scenes with CoordConv offer geometric translation that was never observed in regular GAN. Adding CoordConv to a Faster R-CNN produces much better object boxes and scores. Finally, agents learning to

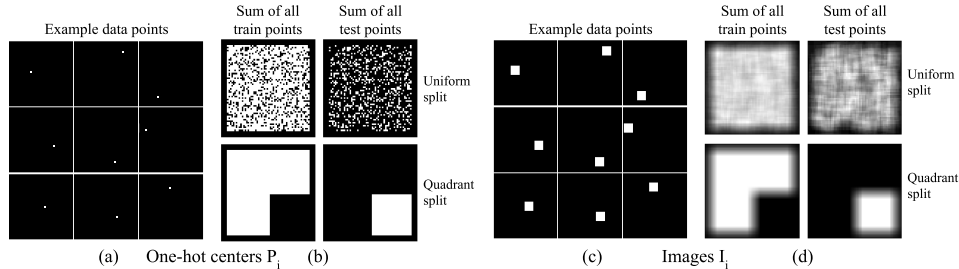

Figure 2: The Not-so-Clevr dataset. **(a)** Example one-hot center images $P_i$ from the dataset. **(b)** The pixelwise sum of the entire train and test splits for uniform vs. quadrant splits. **(c)** and **(d)** Analagous depictions of the canvas images $I_i$ from the dataset. Best viewed electronically with zoom.

      play Atari games obtain significantly higher scores on some but not all games, and they never do significantly worse (Section 5).

6. To enable other researchers to reproduce experiments in this paper, and benefit from using CoordConv as a simple drop-in replacement of the convolution layer in their models, we release our code at `https://github.com/uber-research/coordconv`.

With reference to the above numbered contributions, the reader may be interested to know that the course of this research originally progressed in the $5 \rightarrow 2$ direction as we debugged why progressively simpler problems continued to elude straightforward modeling. But for ease of presentation, we give results in the $2 \rightarrow 5$ direction. A progression of the toy problems considered is shown in Figure 1.

## 2    Not-so-Clevr dataset

We define the Not-so-Clevr dataset and make use of it for the first experiments in this paper. The dataset is a single-object, grayscale version of Sort-of-CLEVR [29], which itself is a simpler version of the Clevr dataset of rendered 3D shapes [14]. Note that the series of Clevr datasets have been typically used for studies regarding relations and visual question answering, but we here use them for supervised learning and generative models. Not-so-Clevr consists of $9 \times 9$ squares placed on a $64 \times 64$ canvas. Square positions are restricted such that the entire square lies within the $64 \times 64$ grid, so that square centers fall within a slightly smaller possible area of $56 \times 56$. Enumerating these possible center positions results in a dataset with a total of 3,136 examples. For each example square $i$, the dataset contains three fields:

- $C_i \in \mathbb{R}^2$, its center location in $(x, y)$ Cartesian coordinates,
- $P_i \in \mathbb{R}^{64 \times 64}$, a one-hot representation of its center pixel, and
- $I_i \in \mathbb{R}^{64 \times 64}$, the resulting $64 \times 64$ image of the square painted on the canvas.

We define two train/test splits of these 3,136 examples: *uniform*, where all possible center locations are randomly split 80/20 into train vs. test sets, and *quadrant*, where three of four quadrants are in the train set and the fourth quadrant in the test set. Examples from the dataset and both splits are depicted in Figure 2. To emphasize the simplicity of the data, we note that this dataset may be generated in only a line or two of Python using a single convolutional layer with filter size $9 \times 9$ to paint the squares from a one-hot representation.[1]

## 3    The CoordConv layer

The proposed CoordConv layer is a simple extension to the standard convolutional layer. We assume for the rest of the paper the case of two spatial dimensions, though operators in other dimensions follow trivially. Convolutional layers are used in a myriad of applications because they often work well, perhaps due to some combination of three factors: they have relatively few learned parameters, they are fast to compute on modern GPUs, and they learn a function that is translation invariant (a translated input produces a translated output).

```
onehots = np.pad(np.eye(3136).reshape((3136, 56, 56, 1)), ((0,0), (4,4), (4,4), (0,0)), "constant");
images = tf.nn.conv2d(onehots, np.ones((9, 9, 1, 1)), [1]*4, "SAME")
```

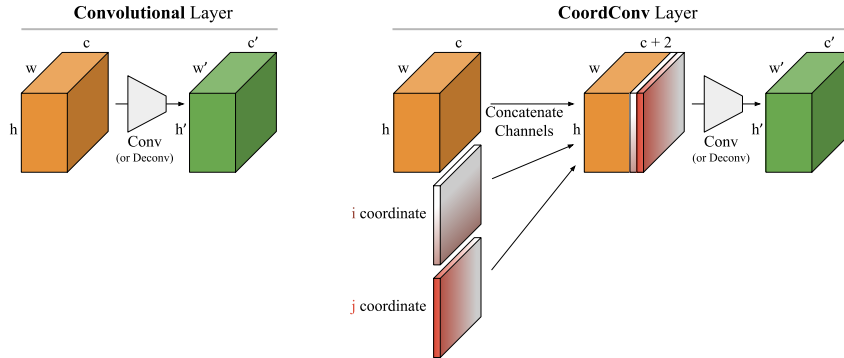

Figure 3: Comparison of 2D convolutional and CoordConv layers. **(left)** A standard convolutional layer maps from a representation block with shape $h \times w \times c$ to a new representation of shape $h' \times w' \times c'$. **(right)** A CoordConv layer has the same functional signature, but accomplishes the mapping by first concatenating extra channels to the incoming representation. These channels contain hard-coded coordinates, the most basic version of which is one channel for the $i$ coordinate and one for the $j$ coordinate, as shown above. Other derived coordinates may be input as well, like the radius coordinate used in ImageNet experiments (Section 5).

The CoordConv layer keeps the first two of these properties—few parameters and efficient computation—but allows the network to learn to keep or to discard the third—translation invariance—as is needed for the task being learned. It may appear that doing away with translation invariance will hamper networks' abilities to learn generalizable functions. However, as we will see in later sections, allocating a small amount of network capacity to model non-translation invariant aspects of a problem can enable far more trainable models that also generalize far better.

The CoordConv layer can be implemented as a simple extension of standard convolution in which extra channels are instantiated and filled with (constant, untrained) coordinate information, after which they are concatenated channel-wise to the input representation and a standard convolutional layer is applied. Figure 3 depicts the operation where two coordinates, $i$ and $j$, are added. Concretely, the $i$ coordinate channel is an $h \times w$ rank-1 matrix with its first row filled with 0's, its second row with 1's, its third with 2's, etc. The $j$ coordinate channel is similar, but with columns filled in with constant values instead of rows. In all experiments, we apply a final linear scaling of both $i$ and $j$ coordinate values to make them fall in the range $[-1, 1]$. For convolution over two dimensions, two $(i, j)$ coordinates are sufficient to completely specify an input pixel, but if desired, further channels can be added as well to bias models toward learning particular solutions. In some of the experiments that follow, we have also used a third channel for an $r$ coordinate, where $r = \sqrt{(i - h/2)^2 + (j - w/2)^2}$. The full implementation of the CoordConv layer is provided in Section S9. Let's consider next a few properties of this operation.

**Number of parameters.** Ignoring bias parameters (which are not changed), a standard convolutional layer with square kernel size $k$ and with $c$ input channels and $c'$ output channels will contain $cc'k^2$ weights, whereas the corresponding CoordConv layer will contain $(c + d)c'k^2$ weights, where $d$ is the number of coordinate dimensions used (e.g. 2 or 3). The relative increase in parameters is small to moderate, depending on the original number of input channels. [2]

**Translation invariance.** CoordConv with weights connected to input coordinates set by initialization or learning to zero will be translation invariant and thus mathematically equivalent to ordinary convolution. If weights are nonzero, the function will contain some degree of translation dependence, the precise form of which will ideally depend on the task being solved. Similar to locally connected layers with unshared weights, CoordConv allows learned translation dependence, but by contrast

it requires far fewer parameters: $(c + d)c'k^2$ vs. $hwcc'k^2$ for spatial input size $h \times w$. Note that all CoordConv weights, even those to coordinates, are shared across all positions, so translation dependence comes only from the specification of coordinates; one consequence is that, as with ordinary convolution but unlike locally connected layers, the operation can be expanded outside the original spatial domain if the appropriate coordinates are extrapolated.

**Relations to other work.** CoordConv layers are related to many other bodies of work. Compositional Pattern Producing Networks (CPPNs) [31] implement functions from coordinates in arbitrarily many dimensions to one or more output values. For example, with two input dimensions and $N$ output values, this can be thought of as painting $N$ separate grayscale pictures. CoordConv can then be thought of as a conditional CPPN where output values depend not only on coordinates but also on incoming data. In one CPPN-derived work [11], researchers did train networks that take as input both coordinates and incoming data for their use case of synthesizing a drum track that could derive both from a time coordinate and from other instruments (input data) and trained using interactive evolution. With respect to that work, we may see CoordConv as a simpler, single-layer mechanism that fits well within the paradigm of training large networks with gradient descent on GPUs. In a similar vein, research on convolutional sequence to sequence learning [7] has used fixed and learned position embeddings at the input layer; in that work, positions were represented via an overcomplete basis that is added to the incoming data rather than being compactly represented and input as separate channels. In some cases using overcomplete sine and cosine bases or learned encodings for locations has seemed to work well [34, 24]. Connections can also be made to mechanisms of spatial attention [13] and to generative models that separately learn what and where to draw [8, 26]. While such works might appear to provide alternative solutions to the problem explored in this paper, in reality, similar coordinate transforms are often embedded within such models (e.g. a spatial transformer network contains a localization network that regresses from an image to a coordinate-based representation [13]) and might also benefit from CoordConv layers.

Moreover, several previous works have found it necessary or useful to inject geometric information to networks, for example, in prior networks to enhance spatial smoothness [32], in segmentation networks [2, 20], and in robotics control through a spatial softmax layer and an expected coordinate layer that map scenes to object locations [18, 5]. However, in those works it is often seen as a minor detail in a larger architecture which is tuned to a specific task and experimental project, and discussions of this necessity are scarce. In contrast, our research (a) examines this necessity in depth as its central thrust, (b) reduces the difficulty to its minimal form (coordinate transform), leading to a simple single explanation that unifies previously disconnected observations, and (c) presents one solution used in various forms by others as a unified layer, easily included anywhere in any convolutional net. Indeed, the wide range of prior works provide strong evidence of the generality of the core coordinate transform problem across domains, suggesting the significant value of a work that systematically explores its impact and collects together these disparate previous references.

Finally, we note that challenges in learning coordinate transformations are not unknown in machine learning, as learning a Cartesian-to-polar coordinate transform forms the basis of the classic two-spirals classification problem [4].

## 4 Supervised Coordinate tasks

### 4.1 Supervised Coordinate Classification

The first and simplest task we consider is Supervised Coordinate Classification. Illustrated at the top of Figure 1, given an $(x, y)$ coordinate as input, a network must learn to paint the correct output pixel. This is simply a multi-class classification problem where each pixel is a class. Why should we study such a toy problem? If we expect to train generative models that can transform high level latents like horizontal and vertical position into pixel data, solving this toy task would seem a simple prerequisite. We later verify that performance on this task does in fact predict performance on larger problems.

In Figure 4 we depict training vs. test accuracy on the task for both uniform and quadrant train/test splits. For convolutional models[3](6 layers of deconvolution with stride 2, see Section S1 in the Supplementary Information for architecture details) on uniform splits, we find models that generalize somewhat, but 100% test accuracy is never achieved, with the best model achieving only 86% test

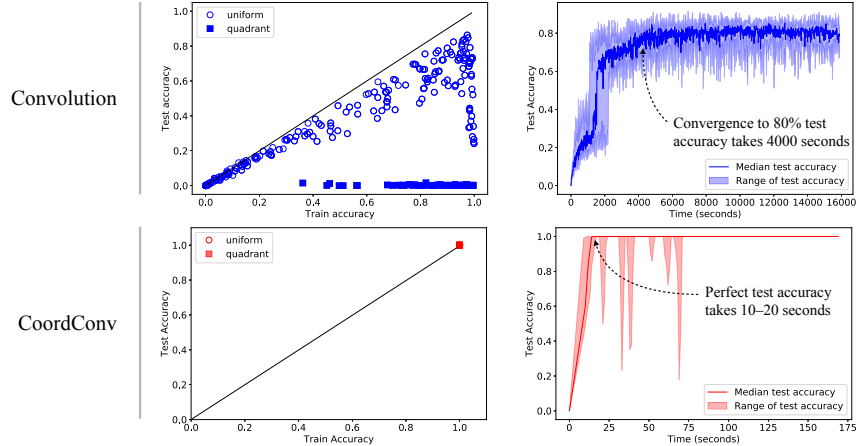

Figure 4: Performance of convolution and CoordConv on Supervised Coordinate Classification. **(left column)** Final test vs. train accuracy. On the easier uniform split, convolution never attains perfect test accuracy, though the largest models memorize the training set. On the quadrant split, generalization is almost zero. CoordConv attains perfect train and test accuracy on both splits. One of the main results of this paper is that the translation invariance in ordinary convolution does not lead to coordinate transform generalization even to neighboring pixels! **(right column)** Test accuracy vs. training time of the best uniform-split models from the left plot (any reaching final test accuracy $\geq 0.8$). The convolution models never achieve more than about 86% accuracy, and training is slow: the fastest learning models still take over an hour to converge. CoordConv models learn several hundred times faster, attaining perfect accuracy in seconds.

accuracy. This is surprising: because of the way the uniform train/test splits were created, all test pixels are close to multiple train pixels. Thus, we reach a first striking conclusion: *learning a smooth function from $(x, y)$ to one-hot pixel is difficult for convolutional networks, even when trained with supervision, and even when supervision is provided on all sides.* Further, training a convolutional model to 86% accuracy takes over an hour and requires about 200k parameters (see Section S2 in the Supplementary Information for details on training). On the quadrant split, convolutional models are unable to generalize at all. Figure 5 shows sums over training set and test set predictions, showing visually both the memorization of the convolutional model and its lack of generalization.

In striking contrast, CoordConv models attain perfect performance on both data splits and do so with only 7.5k parameters and in only 10–20 seconds. The parsimony of parameters further confirms they are simply more appropriate models for the task of coordinate transform [28, 10, 19].

## 4.2 Supervised Coordinate Regression

Because of the surprising difficulty of learning to transform coordinates from Cartesian to a pixel-based, we examine whether the inverse transformation from pixel-based to Cartesian is equally difficult. This is the type of transform that could be employed by a VAE encoder or GAN discriminator to transform pixel information into higher level latents encoding locations of objects in a scene.

We experimented with various convolutional network structures, and found a 4-layer convolutional network with fully connected layers (85k parameters, see Section S3 for details) can fit the uniform training split and generalize well (less than half a pixel error on average), but that same architecture completely fails on the quadrant split. A smaller fully-convolutional architecture (12k parameters, see Section S3) can be tuned to achieve limited generalization on the quadrant split (around five pixels error on average) as shown in Figure 5 (right column), but it performs poorly on the uniform split.

A number of factors may have led to the observed variation of performance, including the use of max-pooling, batch normalization, and fully-connected layers. We have not fully and separately measured how much each factor contributes to poor performance on these tasks; rather we report only that our efforts to find a workable architecture across both splits did not yield any winners. In contrast, a 900 parameter CoordConv model, where a single CoordConv layer is followed by several layers of standard convolution, trains quickly and generalizes in both the uniform and quadrant splits. See Section S3 in Supplementary Information for more details. These results suggest that the inverse transformation requires similar considerations to solve as the Cartesian-to-pixel transformation.

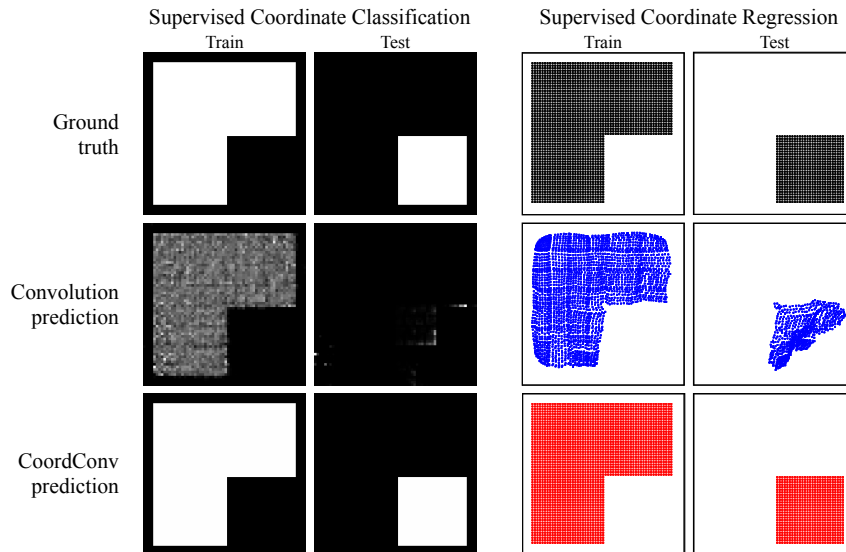

Figure 5: Comparison of convolutional and CoordConv models on the Supervised Coordinate Classification and Regression tasks, on a quadrant split. **(left column)** Results on the seemingly simple classification task where the network must highlight one pixel given its $(x, y)$ coordinates as input. Images depict ground truth or predicted probabilities summed across the entire train or test set and then normalized to make use of the entire black to white image range. Thus, e.g., the top-left image shows the sum of all train set examples. The conv predictions on the train set cover it well, although the amount of noise in predictions hints at the difficulty with which this model eventually attained 99.6% train accuracy by memorization. The conv predictions on the test set are almost entirely incorrect, with two pixel locations capturing the bulk of the probability for all locations in the test set. By contrast, the CoordConv model attains 100% accuracy on both the train and test sets. Models used: conv–6 layers of deconv with strides 2; CoordConv–5 layers of 1×1 conv, first layer is CoordConv. Details in Section S2. **(right column)** The regression task poses the inverse problem: predict real-valued $(x, y)$ coordinates from a one-hot pixel input. As before, the conv model memorizes poorly and largely fails to generalize, while the CoordConv model fits train and test set perfectly. Thus we observe the coordinate transform problem to be difficult in both directions. Models used: conv–9-layer fully-convolution with global pooling; CoordConv–5 layers of conv with global pooling, first layer is CoordConv. Details in Section S3.

## 4.3 Supervised Rendering

Moving beyond the domain of single pixel coordinate transforms, we compare performance of convolutional vs. CoordConv networks on the Supervised Rendering task, which requires a network to produce a $64 \times 64$ image with a square painted centered at the given $(x, y)$ location. As shown in Figure 6, we observe the same stark contrast between convolution and CoordConv. Architectures used for both models can be seen in Section S1 in the Supplementary Information, along with further plots, details of training, and hyperparameter sweeps given in Section S4.

## 5 Applicability to Image Classification, Object Detection, Generative Modeling, and Reinforcement Learning

Given the starkly contrasting results above, it is natural to ask how much the demonstrated inability of convolution at coordinate transforms infects other tasks. Does the coordinate transform hurdle persist insidiously inside other tasks, subtly hampering performance from within? Or do networks skirt the issue by learning around it, perhaps by representing space differently, e.g. via non-Cartesian representations like grid cells [1, 6, 3]? A complete answer to this question is beyond the scope of this paper, but encouraging preliminary evidence shows that swapping Conv for CoordConv can improve a diverse set of models — including ResNet-50, Faster R-CNN, GANs, VAEs, and RL models.

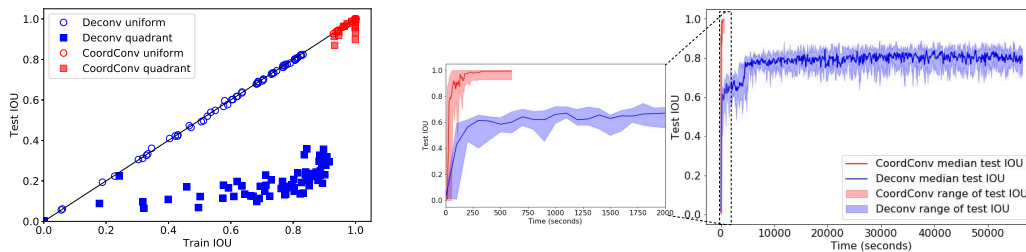

Figure 6: Results on the Supervised Rendering task. As with the Supervised Coordinate Classification and Regression tasks, we see the same vast separation in training time and generalization between convolution models and CoordConv models. **(left)** Test intersection over union (IOU) vs Train IOU. We show all attempted models on the uniform and quadrant splits, including some CoordConv models whose hyperparameter selections led to worse than perfect performance. **(right)** Test IOU vs. training time of the best uniform-split models from the left plot (any reaching final test IOU $\geq 0.8$). Convolution models never achieve more than about IOU 0.83, and training is slow: the fastest learning models still take over two hours to converge vs. about a minute for CoordConv models.

**ImageNet Classification** As might be expected for tasks requiring straightforward translation *invariance*, CoordConv does not help significantly when tested with image classification. Adding a single extra $1\times1$ CoordConv layer with 8 output channels improves ResNet-50 [9] Top-5 accuracy by a meager 0.04% averaged over five runs for each treatment; however, this difference is not statistically significant. It is at least reassuring that CoordConv doesn't hurt the performance since it can always learn to ignore coordinates. This result was obtained using distributed training on 100 GPUs with Horovod [30]; see Section S5 in Supplementary Information for more details.

**Object Detection** In object detection, models look at pixel space and output bounding boxes in Cartesian space. This creates a natural coordinate transform problem which makes CoordConv seemingly a natural fit. On a simple problem of detecting MNIST digits scattered on a canvas, we found the test intersection-over-union (IOU) of a Faster R-CNN network improved by 24% when using CoordConv. See Section S6 in Supplementary Information for details.

**Generative Modeling** Well-trained generative models can generate visually compelling images [23, 15, 36], but careful inspection can reveal mode collapse: images are of an attractive quality, but sample diversity is far less than diversity present in the dataset. Mode collapse can occur in many dimensions, including those having to do with content, style, or position of components of a scene. We hypothesize that mode collapse of position may be due to the difficulty of learning straightforward transforms from a high-level latent space containing coordinate information to pixel space and that using CoordConv could help. First we investigate a simple task of generating colored shapes with, in particular, all possible geometric locations, using both GANs and VAEs. Then we scale up the problem to Large-scale Scene Understanding (LSUN) [35] bedroom scenes with DCGAN [25], through distributed training using Horovod [30].

Using GANs to generate simple colored objects, Figure 7a-d show sampled images and model collapse analyses. We observe that a convolutional GAN exhibits collapse of a two-dimensional distribution to a one-dimensional manifold. The corresponding CoordConv GAN model generates objects that better cover the 2D Cartesian space while using 7% of the parameters of the conv GAN. Details of the dataset and training can be seen in Section S7.1 in the Supplementary Information. A similar story with VAEs is discussed in Section S7.2.

With LSUN, samples are shown in Figure 7e, and more in Section S7.3 in the Supplementary Information. We observe (1) qualitatively comparable samples when drawing randomly from each model, and (2) geometric translating behavior during latent space interpolation.

Latent space interpolation[4] demonstrates that in generating colored objects, motions through latent space generate coordinated object motion. In LSUN, while with convolution we see frozen objects fading in and out, with CoordConv, we instead see smooth geometric transformations including translation and deformation.

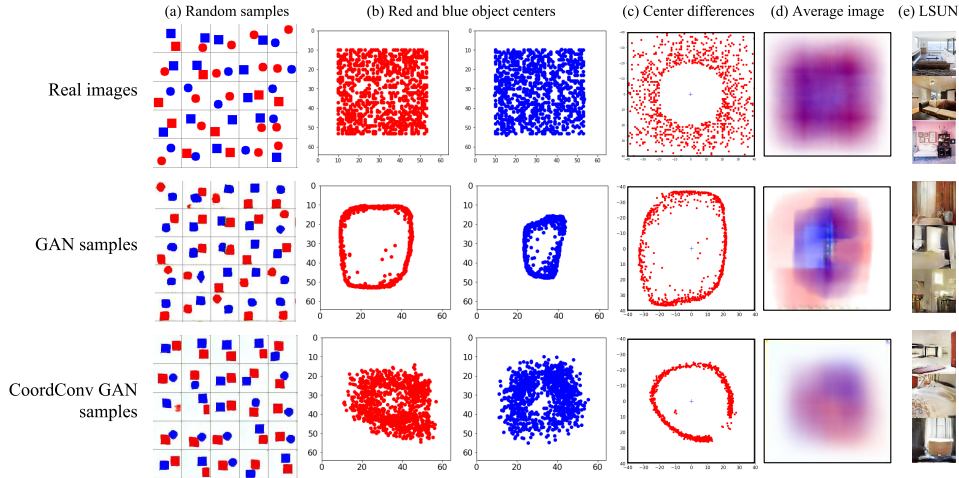

Figure 7: Real images and generated images by GAN and CoordConv GAN. Both models learn the basic concepts similarly well: two objects per image, one red and one blue, their size is fixed, and their positions can be random **(a)**. However, plotting the spread of object centers over 1000 samples, we see that CoordConv GAN samples cover the space significantly better (average entropy: *Data* red 4.0, blue 4.0, diff 3.5; *GAN* red 3.13, blue 2.69, diff 2.81; *CoordConv GAN* red 3.30, blue 2.93, diff 2.62), while GAN samples exhibit mode collapse on where objects can be **(b)**. In terms of relative locations between the two objects, both model exhibit a certain level of model collapse, CoordConv is worse **(c)**. The averaged image of CoordConv GAN is smoother and closer to that of data **(d)**. With LSUN, sampled images are shown **(e)**. All models used in generation are the best out of many runs.

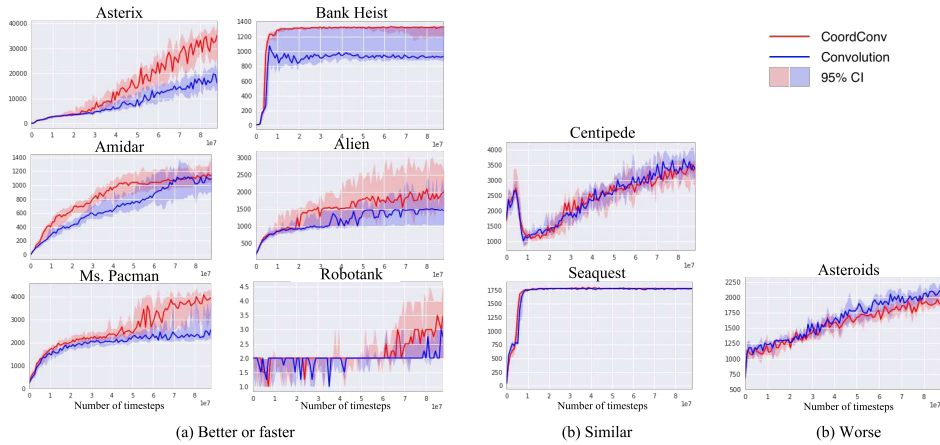

Figure 8: Results using A2C to train on Atari games. Out of 9 games, (a) in 6 CoordConv improves over convolution, (b) in 2 performs similarly, and (c) on 1 it is slightly worse.

**Reinforcement Learning**   Adding a CoordConv layer to an actor network within A2C [22] produces significant improvements on some games, but not all, as shown in Figure 8. We also tried adding CoordConv to our own implementation of Distributed Prioritized Experience Replay (Ape-X) [12], but we did not notice any immediate difference. Details of training are included in Section S8.

## 6   Conclusions and Future Work

We have shown the curious inability of CNNs to model the coordinate transform task, shown a simple fix in the form of the CoordConv layer, and given results that suggest including these layers can boost performance in a wide range of applications. Future work will further evaluate the benefits of CoordConv in large-scale datasets, exploring its ability against perturbations of translation, its impact in relational reasoning [29], language tasks, video prediction, with spatial transformer networks [13], and with cutting-edge generative models [8].

## Acknowledgements

The authors gratefully acknowledge Zoubin Ghahramani, Peter Dayan, and Ken Stanley for insightful discussions. We are also grateful to the entire Opus team and Machine Learning Platform team inside Uber for providing our computing platform and for technical support.

## Footnotes

[1]For example, ignoring import lines and train/test splits:

[2] A CoordConv layer implemented via the channel concatenation discussed entails an increase of $dc'k^2$ weights. However, if $k > 1$, not all $k^2$ connections from coordinates to each output unit are necessary, as spatially neighboring coordinates do not provide new information. Thus, if one cares acutely about minimizing the number of parameters and operations, a $k \times k$ conv may be applied to the input data and a $1 \times 1$ conv to the coordinates, then the results added. In this paper we have used the simpler, if marginally inefficient, channel concatenation version that applies a single convolution to both input data and coordinates. However, almost all experiments use $1 \times 1$ filters with CoordConv.

[3]For classification, convolutions and CoordConvs are actually deconvolutional on certain layers when resolutions must be expanded, but we refer to the models as conv or CoordConv for simplicity.

[4]https://www.youtube.com/watch?v=YefMbLqS7Jg

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
