[Supplementary Material]

# Supplementary Information for:
# An intriguing failing of convolutional neural networks and the CoordConv solution

## S1 Architectures used for supervised painting tasks

Figure S1 depicts architectures used in each of the two supervised tasks going from coordinates to images: Supervised Coordinate Classification (Section 4.1), and Supervised Rendering (Section 4.3).

In the case of convolution, or, in this case, transposed convolution (deconvolution), the same architecture is used for both tasks, as shown in the top row of Figure S1, but we generally found the Supervised Rendering tasks requires wider layers (more channels). Top performing deconvolutional models in Supervised Coordinate Classification have $c = 1$ or $2$, while in Supervised Rendering we usually need $c = 2, 3$. In terms of convolutional filter size, filter sizes of 2 and 4 seem to outperform 3 in Coordinate Classification, while in Rendering the difference is less distinctive.

Note that the CoordConv model only replaces the first layer with CoordConv (shown in green in Figure S1 ).

Figure S1: Deconvolutional and CoordConv architectures used in each of the two supervised tasks. "fs" stands for filter size, and "c" for channel size. We use a grid search on different ranges of them as displayed underneath each model, while allowing deconvolutional models a wider range in both. Green indicates a CoordConv layer.

Figure S2: Model size vs. test accuracy for the Supervised Coordinate Classification subtask on the uniform split and quadrant split. Deconv models (blue) of many sizes achieve 80% or a little higher — but never perfect — test accuracy on the uniform split. On the quadrant split, while many models perform slightly better than chance (1/4096 = .000244) no model generalizes significantly. CoordConv model achieves perfect accuracy on both splits.

Because of the usage of different filter sizes and channel sizes, we end up training models with a range of sizes. Each is combined with further grid searches on hyperparameters including the learning rate, weight decay, and minibatch sizes. Therefore at the same size we end up with multiple models with a spread of performances, as shown in Figure S2 for the Supervised Coordinate Classification task. We repeat the same exact setting of experiments on both uniform and quadrant splits, which result in the same number of experiments. It is not obviously shown in Figure S2 because quadrant trainings are mostly poorly (at the bottom of the figure).

As can be seen, it seems unlikely that even larger models would perform better. They all basically struggle to get to a good test accuracy. This (1) confirms that performance is not simply being limited by model size, as well as (2) shows that working CoordConv models are one to two orders of magnitude smaller (7553 as opposed to 50k-1.6M parameters) than the best convolutional models. The model size vs. test performance plot on Supervised Rendering is similar (not shown), except CoordConv model in that case has a slightly larger number of parameters: 9490. CoordConv achieves perfect test IOU there while deconvolutional models struggle at sizes 183k to 1.6M.

## S2 Further Supervised Coordinate Classification details

For deconvolutional models, we use the model structure as depicted in the top row in Figure S1, while varying the choice of filter size ({2, 3, 4}) and channel size multipliers ({1,2,3}), and each combined with a hyperparameter sweep of learning rate {0.001, 0.002, 0.005, 0.01, 0.02, 0.05}, and weight decay {0.001, 0.01}. Models are trained using a softmax output with cross entropy loss with Adam optimizer. We train 1000 epochs with minibatch size of 16 and 32. The learning rate is dropped to 10% every 200 epochs for four times.

For CoordConv models, because it converges so fast and easy, we did not have to try a lot of settings — only 3 learning rates {0.01 0.001, 0.005} and they all learned perfectly well. There's also no need for learning rate schedules as it quickly converges in 10 seconds.

Figure S3 demonstrates how accurate and smooth the learned probability mass is with CoordConv, and not so much with Deconv. We first show the overall $64 \times 64$ map of logits, one for a training example and one for a test example just right next to the train. Then we zoom in to a smaller region to examine the intricacies. We can see that convolution, even though designed to act in a translation-invariant way, shows artifacts of not being able to accomplish so.

Figure S3: Comparison of behaviors between Deconv model and CoordConv model on the Supervised Coordinate Classification task. We select five horizontally neighboring pixels, containing samples in both train and test splits, and zoom in on a $5 \times 9$ section of the $64 \times 64$ canvas so the detail of the logits and predicted probabilities may be seen. The full $64 \times 64$ map of logits of the first two samples (first in train, second in test) are also shown. The deconvolutional model outputs probabilities in a decidedly non-translation-invariant manner.

## S3    Further Supervised Coordinate Regression details

Exact architectures applied in the Supervised Coordinate Regression task are described in table S1. For the uniform split, the best-generalizing convolution architecture consisted of a stack of alternating convolution and max-pooling layers, followed by a fully-connected layer and an output layer. This architecture was fairly robust to changes in hyperparameters. In contrast, for the quadrant split, the best-generalizing network consisted of strided convolutions feeding into a global-pooling output layer, and good performance was delicate. In particular, training and generalization was sensitive to the number of batch normalization layers (2), weight decay strength (5e-4), and optimizer (Adam, learning rate 5e-4). A single CoordConv architecture generalized perfectly with the same hyperparameters over both splits, and consisted of a single CoordConv layer followed by additional layers of convolution, feeding into a global pooling output layer.

Table S1: Model Architectures for Supervised Coordinate Regression. FC: Fully-connected, MP: Max Pooling, GP: Global Pooling, BN: Batch normalization, s2: stride 2.

|  | Conv | CoordConv |
|---|---|---|
| Uniform Split | 3×3, 16 - MP 2×2 - 3×3, 16 - MP 2×2 - 3×3, 16 - MP 2×2 - 3×3, 16 - FC 64 - FC 2 | 1×1, 8 - 1×1, 8 - 1×1, 8 - 3×3, 8 - 3×3, 2 - GP |
| Quadrant Split | 5×5 (s2), 16 - 1×1, 16 - BN - 3×3, 16 - 3×3 (s2), 16 - 3×3 (s2), 16 - BN - 3×3 (s2), 16 - 1×1, 16 - 3×3 (s2), 16 - 3×3, 2 - GP | |

## S4    Further Supervised Rendering details

Both the architectural and experimental settings are similar to Section S2 except the loss used is pixelwise sigmoid output with cross entropy loss. We also tried mean squared error loss but the performance is even weaker. We performed heavy hyperparameter sweeping and deliberate learning rate annealing for Deconv models (same as said in Section S2), while in CoordConv models it is fairly easy to find a good setting. All models trained with learning rates {0.001, 0.005}, weight decay {0, 0.001}, filter size {1, 2} turned out to perform well after 1–2 minutes of training. Take the best model

obtained, Figure S4 and Figure S5 show the learned logits and pixelwise probability distributions for three samples each, in the uniform and quadrant cases, respectively. We can see that the CoordConv model learns a much smoother and precise distribution. All samples are test samples.

Figure S4: Output comparison between Deconv and CoordConv models on three test samples. Models are trained on a *uniform* split. Logits are model's direct output; pixelwise probability (pw-prob) is logits after Sigmoid. Conv outputs (middle columns) manage to get roughly right. CoordConv outputs (right columns) are precisely correct and its logit maps are smooth.

Figure S5: Output comparison between Deconv and CoordConv models on three test samples. Models are trained on a *quadrant* split. Logits are model's direct output; pixelwise probability (pw-prob) is logits after Sigmoid. Conv outputs (middle columns) failed mostly. Even with such a difficult generalization problem, CoordConv outputs (right columns) are precisely correct and its logit maps are smooth.

## S5 Further ImageNet classification details

We evaluate the potential of CoordConv in image classification with ImageNet experiments. We take ResNet-50 and run the baseline on distributed framework using 100 GPUs, with the open-source framework Horovod. For CoordConv variants, we add an extra CoordConv layer only in the beginning, which takes a 6-channel tensor containing image RBG, $i$, $j$ coordinates and pixel distance to center $r$, and output 8 channels with $1\times1$ convolution. The increase of parameters is negligible. It then goes in with the rest of ResNet-50.

Each model is run 5 times on the same setting to account for experimental variances. Table. S2 lists the test result from each run in the end of 90 epochs. CoordConv model obtains better average result on two of the three measures, however a one-sided t-test tells that the improvement on Top 5 accuracy is not quite statistically significant with $p = .11$.

Of all vision tasks, we might expect image classification to show the least performance change when using CoordConv instead of convolution, as classification is more about what is in the image than where it is. This tiny amount of improvement validates that.

Table S2: ImageNet classification result comparison between a baseline ResNet-50 and CoordConv ResNet-50. For each model three experiments are run, listed in three separate rows below.

|  | Test loss | Top-1 Accuracy | Top-5 Accuracy |
|---|---|---|---|
| Baseline ResNet-50 | 1.43005 | 0.75722 | 0.92622 |
|  | 1.42385 | 0.75844 | 0.9272 |
|  | 1.42634 | 0.75782 | 0.92754 |
|  | 1.42166 | 0.75692 | 0.92756 |
|  | 1.42671 | 0.75724 | 0.92708 |
| Average | 1.425722 | **0.757528** | 0.92712 |
| CoordConv ResNet-50 | 1.42335 | 0.75732 | 0.92802 |
|  | 1.42492 | 0.75836 | 0.92754 |
|  | 1.42478 | 0.75774 | 0.92818 |
|  | 1.42882 | 0.75702 | 0.92694 |
|  | 1.42438 | 0.75668 | 0.92714 |
| Average | **1.42525** | 0.757424 | **0.927564** |

## S6  Further object detection details

The object detection experiments are on a dataset containing randomly rescaled and placed MNIST digits on a $64 \times 64$ canvas. To make it more akin to natural images, we generate a much larger canvas and then center crop it to be $64 \times 64$, so that digits can be partially outside of the canvas. We kept images that contain 5 digit objects whose centers are within the canvas. In the end we use 9000 images as the training set and 1000 as test.

A schematic of the model architecture is illustrated in Figure S6. We use number of anchors $A = 9$, with sizes $(15, 15)$, $(20, 20)$, $(25, 25)$, $(15, 20)$, $(20, 25)$, $(20, 15)$, $(25, 20)$, $(15, 25)$, $(25, 15)$. In box sampling (training mode), $p\_size$ and $n\_size$ are 6. In box non-maximum suppression (NMS) (test mode), the IOU threshold is 0.8 and maximum number of proposal boxes is 10. After the boxes are proposed and shifted, we do not have a downstream classification task, but just calculate the loss from the boxes. The training loss include box loss and score loss. As evaluation metric we also calculate IOUs between proposed boxes and ground truth boxes. Table. S3 lists those metrics obtained the test dataset, by both Conv and CoordConv models. We found that every metric is improved by CoordConv, and the average test IOU improved by about 24 percent.

Figure S6: Faster R-CNN architecture used for object detection on scattered MNIST digits. Green indicates where coordinates are added. Note that the input image is used for demonstration purpose. The real dataset contains 5 digits on a canvas and allows overlapping. **(Left)** train mode with box sampler. **(Right)** test mode with box NMS.

Table S3: MNIST digits detection result comparison between a Faster R-CNN model with regular convolution vs. with CoordConv. Metrics are all on test set. Train IOU: average IOU between sampled positive boxes (train mode) and ground truth; Test IOU-average): average IOU between 10 selected boxes (test mode) and ground truth; Test IOU-select: average IOU between the best scored box and its closest ground truth.

|                                  | Conv   | CoordConv | % Improvement |
|----------------------------------|--------|-----------|---------------|
| Box loss                         | 0.1003 | 0.0854    | 17.44         |
| Score loss                       | 0.5270 | 0.2526    | 108.63        |
| Total loss (sum of the two above)| 0.6271 | 0.3383    | 85.37         |
| Train IOU                        | 0.6388 | 0.6612    | 3.38          |
| Test IOU-average                 | 0.1508 | 0.1868    | 23.87         |
| Test IOU-select                  | 0.4965 | 0.6359    | 28.08         |

## S7   Further generative modeling details

### S7.1   GANs on colored shapes

**Data.**   The dataset used to train the generative models is 50k red-and-blue-object images of size $64 \times 64$. We follow the same mechanism as Sort-of-Clevr, in that objects appear at random positions on a white background without overlapping, only limiting the number of objects to be 2 per image. The objects are always one red and one blue, of a randomly chosen shape out of {circle, square}. Examples of images from this dataset can be seen in the top row, leftmost column in Figure 7, at the intersection of "Real images" and "Random samples".

**Architecture and training details.**   The $z$ dimension to both regular GAN and CoordConv GAN is 256. In GAN, the generator uses 4 layers of deconvolution with strides of 2 to project $z$ to a $64 \times 64$ image shape. The parameter size of the generator is 6,413,315. In CoordConv GAN, we add coordinate channels only at the beginning, making the first layer CoordConv, and then continue with normal Conv.. The generator in this case uses mostly (1,1) convolutions and has only 444,931 parameters. The same discriminator is used for both models. In the case where we also turn the

discriminator to be CoordConv like, its first Conv layer is replaced by CoordConv, and the parameter size increases from 4,316,545 to 4,319,745. The details of both architectures can be seen in Table. S4. We trained two CoordConv GAN versions where CoordConv applies: 1) only in generator, and 2) in both generator and discriminator. They end up performing similarly well. The demonstrated examples in all figures are from one in the latter case.

The change needed to make a generator whose first layer is fully-connected CoordConv is trivial. Instead of taking image tensors which already have Cartesian dimensions, the CoordConv generator first tiles $z$ vector into a full $64 \times 64$ space, and then concatenate it with coordinates in that space.

To train each model we use a fixed learning rate 0.0001 for the discriminator and 0.0005 for the generator. In each iteration discriminator is trained once followed by generator trained twice. The random noise vector $z$ is drawn from a uniform distribution between $[-1, 1]$. We train each model for 50 epochs and save the model in the end of every epoch. We repeat the training with the same hyperparameters 5 to 10 times for each, and pick the best model for each to show a fair comparison in all figures.

Table S4: Model Architectures for GAN and CoordConv GAN used in the colored shape generation. In the case of CoordConv GAN, only the first layer is changed from regular Conv to CoordConv. FC: fully connected layer; s2: stride 2.

|  | Generator | Discriminator |
|---|---|---|
| GAN | FC 8192 (reshape 4×4×512) - 5×5, 256 (s2) - 5×5, 128 (s2) - 5×5, 64 (s2) - 5×5, 3 (s2) - Tanh | 5×5, 64 (s2) - 5×5, 128 (s2) - 5×5, 256 (s2) - 5×5, 512 (s2) - 1 |
| CoordConv GAN | 1×1, 512 - 1×1, 256 - 1×1, 256 - 1×1, 128 - 1×1, 64 - 3×3, 64 - 3×3, 64 - 1×1, 3 | |

**Latent interpolation.** Latent interpolation is conducted by randomly choosing two noise vectors, each from a uniform distribution, and linearly interpolate in between with an $\alpha$ factor that indicates how close it is to the first vector. Figure S7 and Figure S8 each show, on regular GAN and CoordConv GAN, respectively, five random samples of pairs to conduct interpolation with. In addition to Figure S8, Figure S9 shows deliberately picked examples that exhibit a special moving effect that has only been seen in CoordConv GAN.

Figure S7: Regular GAN samples with a series of interpolation between 5 random pairs of $z$. Also observed position and shape transitioning but are different.

**Measure of entropy.** In Figure 7, we reduce generated red and blue objects to their centers and plot the coverage of space in column (b) and relative locations in (c). To make the comparison quantitative,

Figure S8: CoordConv GAN samples with a series of interpolation between 5 random pairs of $z$. Top row: at the position in the manifold, the model has learned a smooth circular motion. The rest of the rows: the circular relation between two objets is still observed, while some object shapes also undergo a smooth transition.

Figure S9: A special effect only observed in CoordConv GAN latent space interpolation: two objects stay constant in relative positions to each other but move together in space. They even move out of the scene which is never present in the real data — learned to extrapolate. These 3 examples are picked from many random drawings of $z$ pairs, as opposed to Figure S8 and Figure S7, where first 5 random drawings are shown.

we can further calculate the entropy in each case, reducing each figure in (b) and (c) to an entropy value shown as a bar in Figure S10. Confidence intervals of each bar is also shown by repeating the experiment 10 times. We can see that CoordConv (red) is closer to data (green) in objects' coverage of space, but has more of a mode collapse in objects' relative position.

## S7.2   VAEs on colored shapes

We train both convolutional and CoordConv VAEs on the same dataset of 50k 64 x 64 images of blue and red non-overlapping squares and circles, as described in Section S7.1. Convolutional VAEs exhibit many of the same problems that we observed in GANs, and adding CoordConv confers many of the same benefits.

A VAE is composed of an encoder that maps data to a latent and a decoder that maps the latent back to data. With minor exceptions our VAE's encoder architecture is the same as our GAN's discriminator and it's decoder is the same as our GAN's generator. The important difference is of course that the output shape of the encoder is the size of the latent (in this case 8), not two as in a discriminator.

Figure S10: Entropy values and confidence intervals of the sampled results in Figure 7, column (b) and (c).

Architectures are shown in Table. S5. The decoder architectures of the convolutional control and CoordConv experiments are similar aside from kernel size - the CoordConv decoder uses 1x1 kernels while the convolutional decoder uses 5x5 kernels.

Due to the pixel sparsity of images in the dataset we found it important to weight reconstruction loss more heavily than latent loss by a factor of 50. Doing so didn't interfere with the quality of the encoding. We used Adam with a learning rate of 0.005 and no weight decay.

Table S5: Model Architectures: Convolutional VAE and CoordConv VAE

|  | Decoder | Encoder |
|---|---|---|
| VAE | FC 8192 (reshape $4\times4\times512$) - $5\times5$, 256 (s2) - $5\times5$, 128 (s2) - $5\times5$, 64 (s2) - $5\times5$, 3 (s2) - Sigmoid | $5\times5$, 64 (s2) - $5\times5$, 128 (s2) - $5\times5$, 256 (s2) - $5\times5$, 512 (s2) - Flatten - FC, 10 |
| CoordConv VAE | $1\times1$, 512 - $1\times1$, 256 - $1\times1$, 256 - $1\times1$, 128 - $1\times1$, 64 - $1\times1$, 3 - Sigmoid | |

Figure S11: Latent space interpolations from a VAE without CoordConv. The red and blue shapes are mostly stationary. When they do move they do so by disappearing and appearing elsewhere in pixel space. Smooth changes in the latent don't translate to smooth geometric changes in pixel space. The latents we interpolated between were sampled randomly from a uniform distribution.

Figure S12: Latent space interpolations from a VAE with CoordConv in the encoder and decoder. The red and blue shapes span pixel space more fully and smooth changes in latent space map to smooth changes in pixel space. Like the CoordConv GAN, the CoordConv VAE is able to extrapolate beyond the borders of the frame it was trained on. The latents we interpolated between were sampled randomly from a uniform distribution.

## S7.3   GANs on LSUN

The dataset used to train the generative models is LSUN bedroom, composed of 3,033,042 images of size $64 \times 64$.

The architectures adopted (see Table. S6) are similar to the ones adopted for generating the colored shape results in Section S7.1, with a few noticeable differences:

- We use CoordConv layers instead of regular Conv layers not only in the first layer of the discriminator, but in each layer. $z$ is of dimension 100.

- The GAN generator includes a layer mapping from $z$ to a 4x4x1024 tensor and the other layers have double the number of channels.

- CoordConv GAN generator has more channels for each layer.

Table S6: Model Architectures for GAN and CoordConv GAN for LSUN. FC: fully connected layer; s2: stride 2.

| | Generator | Discriminator |
|---|---|---|
| GAN | FC 16384 (reshape $4\times4\times1024$) - $5\times5$, 512 (s2) - $5\times5$, 256 (s2) - $5\times5$, 128 (s2) - $5\times5$, 3 (s2) - Tanh | $5\times5$, 64 (s2) - $5\times5$, 128 (s2) - $5\times5$, 256 (s2) - $5\times5$, 512 (s2) - 1 |
| CoordConv GAN | $1\times1$, 1024 - $1\times1$, 512 - $1\times1$, 256 - $1\times1$, 256 - $1\times1$, 128 - $3\times3$, 128 - $3\times3$, 64 - $1\times1$, 3 | |

<div align="center">GAN          CoordConv GAN</div>

Figure S13: Samples from the regular GAN (left) and the CoordConv GAN (right).

Samples from both models are provided in Figure S13. One peculiar property of the CoordConv GAN model with respect to the regular GAN one is the geometric interpolation. As shown in Figure Figure S14 in regular GAN interpolations objects appear and disappear, while in CoordConv GAN interpolations in Figure S15 objects move around, translating, enlarging, squashing and doing geometric transformations over them.

Figure S14: Samples of regular GAN trained on LSUN with a series of interpolation between 5 random pairs of $z$.

Figure S15: Samples of CoordConv GAN trained on LSUN with a series of interpolation between 5 random pairs of $z$.

The regular GAN has been trained for 11000 steps of batch size 128, while the CoordConv GAN has been trained 22000 steps of batch size 64 (because the available memory on the GPUs did not allow for 128). Both models have been trained using Horovod to distribute the training on 16 GPUs.

## S8 Further reinforcement learning details

We used OpenAI baselines [5] implementation and default parameters on all experiments. Table. S7 shows the average scores obtained at the end of game over 10 runs of each.

Table S7: All games with final scores and p-values.

| Game | Conv | CoordConv | p-value |
|---|---|---|---|
| Alien | 1462.5 | 2005.0 | 0.0821 |
| Bank Heist | 932.5 | 1330.0 | 0.1736 |
| Ms. Pacman | 2557.5 | 3945.0 | 0.0065 |
| Robotank | 2.75 | 3.5 | 0.2899 |
| Centipede | 3359.5 | 3424.5 | 0.8703 |
| Asterix | 16250.0 | 35300.0 | 0.0003 |
| Asteroids | 2082.5 | 1912.5 | 0.1124 |
| Amidar | 1092.75 | 1137.5 | 0.2265 |
| Seaquest | 1780.0 | 1780.0 | 0.4057 |

## S9 The CoordConv layer implementation

```
from tensorflow.python.layers import base
import tensorflow as tf

class AddCoords(base.Layer):
    """Add coords to a tensor"""
    def __init__(self, x_dim=64, y_dim=64, with_r=False):
        super(AddCoords, self).__init__()
        self.x_dim = x_dim
        self.y_dim = y_dim
        self.with_r = with_r

    def call(self, input_tensor):
        """
        input_tensor: (batch, x_dim, y_dim, c)
        """
        batch_size_tensor = tf.shape(input_tensor)[0]
        xx_ones = tf.ones([batch_size_tensor, self.x_dim],
                    dtype=tf.int32)
        xx_ones = tf.expand_dims(xx_ones, -1)
        xx_range = tf.tile(tf.expand_dims(tf.range(self.y_dim), 0),
                    [batch_size_tensor, 1])
        xx_range = tf.expand_dims(xx_range, 1)

        xx_channel = tf.matmul(xx_ones, xx_range)
        xx_channel = tf.expand_dims(xx_channel, -1)

        yy_ones = tf.ones([batch_size_tensor, self.y_dim],
                    dtype=tf.int32)
        yy_ones = tf.expand_dims(yy_ones, 1)
        yy_range = tf.tile(tf.expand_dims(tf.range(self.x_dim), 0),
                    [batch_size_tensor, 1])
```

```
        yy_range = tf.expand_dims(yy_range, -1)

        yy_channel = tf.matmul(yy_range, yy_ones)
        yy_channel = tf.expand_dims(yy_channel, -1)

        xx_channel = tf.cast(xx_channel, 'float32') / (self.x_dim - 1)
        yy_channel = tf.cast(yy_channel, 'float32') / (self.y_dim - 1)
        xx_channel = xx_channel*2 - 1
        yy_channel = yy_channel*2 - 1

        ret = tf.concat([input_tensor,
                        xx_channel,
                        yy_channel], axis=-1)

        if self.with_r:
            rr = tf.sqrt( tf.square(xx_channel)
                    + tf.square(yy_channel)
                    )
            ret = tf.concat([ret, rr], axis=-1)

        return ret

class CoordConv(base.Layer):
    """CoordConv layer as in the paper."""
    def __init__(self, x_dim, y_dim, with_r, *args, **kwargs):
        super(CoordConv, self).__init__()
        self.addcoords = AddCoords(x_dim=x_dim,
                                    y_dim=y_dim,
                                    with_r=with_r)
        self.conv = tf.layers.Conv2D(*args, **kwargs)

    def call(self, input_tensor):
        ret = self.addcoords(input_tensor)
        ret = self.conv(ret)
        return ret
```

## Footnotes

[5]https://github.com/openai/baselines/