[Reviews · NeurIPS 2018]

Reviewer 1



Update following author response Thank you for addressing many of the concerns. Overall, all my previous concerns about the quality, clarity and significance are alleviated. However I am not very certain about the originality still. ------------------------ - summary of the main ideas It was difficult to provide a concise summary of the main ideas because there is a lot going on in this paper. Note that various models were used for experimentation and that model specifications are given in the supplementary material. It was also difficult to keep track of which kind of convolutional neural networks (CNNs) were used for which subproblem. First, this paper shows instances where CNNs fail at particular subsets of the coordinate transform problem. For "normal" CNNs, the paper shows: 1. that a fully convolutional model fails to solve the supervised coordinate classification problem and the supervised rendering problem. 2. that a convolutional model with max-pooling and fully connected layers fails to solve the supervised coordinate regression problem, specifically the quadrant split. However the respective model generalizes well for the supervised coordinate regression problem with the uniform split. 3. that a fully convolutional model achieves limited generalization on the supervised coordinate regression problem with quadrant split. The respective model fails on the supervised coordinate regression problem with uniform split. Second, a solution called CoordConv is proposed to solve this problem, generalized as the coordinate transform problem. With CoordConv, the input to the CNN is augmented with 2 extra channels representing the x and y Cartesian coordinates respectively. This is the only modification needed to implement CoordConv. They show that using CoordConv, all the above mentioned toy problems are solved with extremely high accuracy, requiring far less training time and utilizing smaller CNNs (less model parameters). Third, to show that the CoordConv solution can increase performance on other problems, and not just this toy problem, a comparison is made between "normal" CNNs and CoordConv CNNs on the following problems: - Unsupervised Density Learning with GANs - Image classification with ResNet-50 - Deep Reinforcement Learning (RL) on a selection of Atari games with the A2C network - quality -- strengths .Performance on the toy problems was demonstrated very well and the results were impressive. .Figure 8: (a) I was surprised to see the great improvement in performance because MsPacman is notoriously difficult. .In the RL experiment and the image classification task, significance testing was performed. -- weaknesses .In lines 22 and 72 it is mentioned that for the ImageNet classification with ResNet-50 the performance increase is significant, with p=0.051. This is simply a false claim, which you seem to realize as well, since you write in line 224 that this result is "nearly statistically significant". .Figure 7: The results of the GAN and CoordConv GAN are ambiguous. For example, I see no difference between the two models in (a) and in (c) I can argue that the CoordConv is the one suffering from mode collapse, since the "normal" GAN shows a more diverse sampling. For (d) I don't know how to interpret this. .No quantifiable measure for mode collapse is used. It would have been much better if an objective measure for mode collapse had been used, such as entropy. .Figure 8: (b) it is highly questionable that this graph is cut off at timestep 7, exactly when the overlap in performance happens. This result seems cherry-picked. Especially since (a) and (c) go up to timestep 8. Please explain why this was done, ideally provide a graph that goes to timestep 8. Also, why is there no table of all the Atari games together with their results? .No theoretical justification was given whatsoever for the behaviors exhibited by the various neural networks. The rigor of the experiments outside of the toy context was not satisfactory to a NIPS standard. .The role of max-pooling for achieving translation invariance is not addressed. For example, no explanation is given for the differences in performance between (2) and (3) in my summary above. To me it is clear that max-pooling will be problematic for tasks that require translation dependence, since max-pooling increases translation invariance. .In line 162 the paper claims to verify that "performance on this task does in fact predict performance on larger problems". However, no such verification is presented. .The reasoning for the demonstration on the RL task and the image classification task is not given. I am left with the question of "why would these specific tasks benefit from CoordConv?". To me it seems that a location specific task would benefit from CoordConv, however with the image classification I don't see the usefulness of CoordConv because ImageNet classification task is translation invariant. - clarity -- strengths .Mostly very good visuals, clearly representing the benefit of CoordConv on the toy problems. .Paper was well structured -- weaknesses .It was difficult to keep track of which kind of convolutional neural networks (CNNs) were used for which subproblem. A table showing an overview of the various models would have been very useful. .The paper does not provide enough information for replication. Even though it is mentioned how the grid search for parameters is performed, it is not clear which final networks were used to obtain the figures. .Figure 7 did not give a clear overview of the differences between the models. .It is not clear to me how to implement the CoordConv solution in the supervised coordinate classification problem and the supervised rendering problem. In Figure S1, what does "32-32-64-64-1" mean? - originality -- strengths .This is the first time that I have seen such an approach to expand the capabilities of a CNN towards translation dependence. -- weaknesses .I am not entirely sure how original this contribution is, given that there is another paper which uses a similar approach of channel coordinate augmentation in their pipeline to perform image inpainting, see Figure 2 in the supplementary material: Ulyanov, Dmitry, Andrea Vedaldi, and Victor Lempitsky. "Deep image prior." arXiv preprint arXiv:1711.10925 (2017). Supplementary material: https://box.skoltech.ru/index.php/s/ib52BOoV58ztuPM#pdfviewer However said paper does not make this the main focus of their work and it is also not clear to me if the "meshgrid" channels are directly added to the input. - significance -- strengths .The failing of traditional CNNs (with max pooling or fully convolutional) has been adequately demonstrated on the toy problems. It is also clear that the CoordConv solution solves these problems. -- weaknesses .However, it is not clear to me that this solution is useful beyond a toy context.

Reviewer 2



Update following the author rebuttal: Thank you to the authors for a thoughtful rebuttal. The updated results, experiments and clarity of details give much more insight into the paper. Though, the solution is not completely unique, I agree with the authors that understanding and doing a systematic study of adding positional encodings across different tasks, is really helpful. _____________________________________ The authors propose a new operation called the CoordConv, that explicitly adds co-ordinate information to the convolution operation to be more robust towards translation invariance and dependence. The solution is adding the (x,y) coordinate representation before doing the conv operation. It is a simple yet effective operation. By adding position information, the CoordConv performs much better and faster on the Not-so-Clevr dataset for both Supervised rendering and Coordinate classification task. The performance gain on ImageNet for the CoordConv op seems really small. Applying the CoordConv solution to MNIST and checking performance again perturbations of translation, scale, rotation, etc will show the op's generalization property. This will also help to compare with Spatial Transformation Networks. There has also been other related work to add position embeddings, either learned or using sine and cosine of different frequencies. So, I am not sure how novel the technique is. https://arxiv.org/pdf/1802.05751.pdf other comments: In the Supervised Coordinate Classification, why does the test accuracy have so much variance?

Reviewer 3



Updated post-rebuttal. I want to thank the authors for addressing many of the concerns. I believe most of them were discussed. Though the method isn’t necessarily novel, the exploration and toy experiments are thorough and compelling and the paper is clear and well written. I agree with the author's statement about the work “leading to a simple explanation that unifies previously disconnected observations”. I think it might be relevant to look at/cite the following papers that also use positional encoding: A simple neural network module for relational reasoning Image Transformer Attention Is All You Need I also think the phrase "CoordConv layer improves Resnet-40 Top-5 accuracy on average by 0.04%..." is still a little weak to include in the paper. I think a clearer explanation would be that it doesn't though it doesn't improve accuracy by a statistically relevant margin but this is not surprising given it is a problem that is helped by translation invariance. Perhaps this even shows that CoordConv doesn't decrease accuracy on translation invariant problems, which is a useful result. I do think the VQA and Clevr extensions would be useful as well as discussions about max pooling and data augmentation with CoordConv but I think that is fair to classify those as future work. --------------------------------------------------------------------------------------------------------- The paper discusses a failing of convolutional networks when doing coordinate transforms. They explore variations of a toy problem generated from the Clevr dataset in order to experiment with this issue. They further test this method with a comparison on the effects on a GAN, RL for game play, and Resnet-50 on imagenet. The authors propose an interesting solution and although it is well explained in the paper, there are many similar methods in VQA papers that have a similar approach as just a small part of their method. The toy evaluation here is very useful and I think a good look into what is happening but there is not much evaluation past that and it is very limited. The effect of data augmentation is fully removed here, which is an important part of how current state of the art methods deal with invariance. It would be interesting to see how much of this effect is mitigated by that. The authors cite [22] “A simple neural network module for relational reasoning”, which discusses a methodology for relational reasoning that is very much related to the proposed method where relative spatial methods are given to the RN as as input with the convolutional feature maps. The authors do not discuss this related work nor compare to it directly. Other methods that do something related to this include “Inferring and Executing Programs for Visual Reasoning” and “Modeling Relationships in Referential Expressions with Compositional Modular Networks”. With the bold claims (“CoordConv allows networks to learn either perfect translation invariance or varying degrees of translation dependence, as required by the end task […] and with perfect generalization.”) the authors have for relational reasoning given in the abstract, why not compare directly with these methods by just adding the CoordConv layer to these existing methods and checking for improvement? Overall, I feel that the work is interesting, very clear, and the toy problem and it’s evaluation are well explained. However, little to no discussion is given to related work and their is little to no evaluation on real datasets. The resnet results are not statistically significant nor are they convincingly better. The Atari experiments are barely discussed with little insight into how the evaluation is done and no comparison with existing work and state-of-the-art. There is also no state-of-the-art comparison with Clevr where locational relevance is an important feature. The authors state: “similar coordinate transforms are often embedded within such models and might also benefit from CoordConv layers”. This is the type of evaluation I think the paper needs to be successful. Test the inclusion of the layer on existing methods on Clevr or other benchmarks to show that the method is successful on more than a toy problem. I think that if they can show this method’s impact on convolutional networks in a much more convincing manner. With the proper evaluation, I think this could be a very strong paper that would fit well at NIPS, however, it is currently missing that. Note on references: [5] and [6] are the same work.